# Differences in Physicochemical Properties of Stems in Oat (*Avena sativa* L.) Varieties with Distinct Lodging Resistance and Their Regulation of Lodging at Different Planting Densities

**DOI:** 10.3390/plants13192739

**Published:** 2024-09-30

**Authors:** Lingling Liu, Guoling Liang, Wenhui Liu

**Affiliations:** Key Laboratory of Superior Forage Germplasm in the Qinghai-Tibetan Plateau, Qinghai Academy of Animal Science and Veterinary Medicine, Qinghai University, Xining 810016, China; liu_ling_60@sina.com (L.L.); qhliuwenhui@163.com (W.L.)

**Keywords:** lodging coefficient, regulation mechanism, forage quality

## Abstract

Planting density is an effective strategy for regulating both *oat* lodging and forage quality. To delve into the regulatory mechanisms of planting density on lodging and *oat* forage quality, lodging-resistant variety LENA and lodging-sensitive variety QY2 were grown in 2018 and 2019 growing seasons, and four planting densities were implemented: 2.25 × 10^6^ plants/ha (D1), 4.5 × 10^6^ plants/ha (D2), 6.75 × 10^6^ plants/ha (D3), and 9 × 10^6^ plants/ha (D4). At the milk stage, we measured the contents of potassium, calcium, magnesium, silicon, lignin, crude fiber, starch, soluble sugar, and soluble protein in the second and third stem internodes of the plants. The results revealed the lodging-resistant variety LENA demonstrated significantly higher contents of calcium, potassium, silicon, crude fiber, lignin, and lower contents of starch, soluble sugar, and soluble protein (*p* < 0.01). Similar trends in the physicochemical properties of stem internodes for both *oat* varieties with increasing planting density. Crude fiber, soluble sugar, magnesium, starch, potassium, and lignin were the key characteristics affecting the lodging coefficient, and variety and planting density affected the lodging coefficient mainly by regulating the synthesis of starch, soluble sugar, and crude fiber. At planting density D3, stem internodes exhibited higher physicochemical properties and a lower lodging coefficient, favoring *oat* forage production. The results offer a valuable theoretical foundation and practical reference for *oat* lodging-resistant cultivation.

## 1. Introduction

In the global context, *oat* is a crucial dual-purpose crop for grain and forage. Firstly, these plants serve as an excellent source of forage, contributing significantly to the constructive development of the livestock industry [1]. Secondly, the grains of *oat* are rich in nutrients, and people are increasingly interested in them as health food [2]. With an increasing profound understanding of the nutritional, ecological, and economic values associated with *oat*, global demand for this versatile crop has been steadily rising. According to statistics, the global *oat* cultivation area reached 9.492 million hectares in 2021, with China accounting for 5.53%, and the cultivation area continues to expand (https://www.huaon.com/channel/trend/892251.html (accessed on 9 May 2024)). Consequently, the development of *oat* industry is of great significance to meet market demands and the development of related industries.

Lodging stands as a critical factor that limits the improvement in *oat* yield and quality and hinders the process of mechanized production [3]. It has been estimated that lodging can result in *oat* yield losses of up to 37–40% [4], and the earlier lodging occurs, the higher the yield losses [5]. Lodging not only causes the deterioration of the microenvironment in the crop population but also results in a reduction in photosynthetic performance of leaves, an increase in diseases and pests, and premature aging, ultimately resulting in severe adverse effects on yield and quality [6]. Therefore, enhancing *oat* lodging resistance is an urgent matter that needs to be addressed.

Currently, some physicochemical characteristics conducive to enhancing crop lodging resistance have been identified. Carbohydrate components such as lignin, cellulose, starch, and soluble sugars are associated with stem stiffness [7]. Welton proposed that lodging in *wheat* results from insufficient lignin in stems [8], with lignin content showing a significant positive correlation with stem breaking resistance and a negative correlation with lodging rate [9,10]. The higher accumulation of cellulose and related enzyme activities can enhance stem breaking strength and lodging resistance [11]. The substantial accumulation of starch also contributes to increased breaking strength and stiffness of stems, thereby enhancing lodging resistance [12,13]. Soluble sugar deficiency inhibits stem growth, leading to reduced internode breaking strength [14]. Mineral elements, including calcium, potassium, magnesium, and silicon, also play important roles in crop lodging resistance. Potassium binds with lignin and cellulose in the cell wall, strengthening *wheat* stems [15]. Adequate calcium content helps strengthen the structure of the plant cell wall [16]. Magnesium supports the synthesis of organic substances [17]. Silicon promotes the silicification and lignification of the cell wall, enhancing lodging resistance [18]. Additionally, soluble protein content is closely related to plant metabolism, contributing significantly to crop stress resistance [19]. Therefore, gaining a deeper understanding of the influencing characteristics of these physicochemical properties is essential for enhancing crop lodging resistance.

The planting density effectively regulates the utilization and allocation of resources within the population, thereby influencing lodging. Studies have shown that high planting density can affect the crop population structure and nutrient absorption, resulting in weak stems and increased lodging [20]. Additionally, it can also weaken the anchoring strength of the roots by reducing root size [21,22]. However, low planting density may lead to resource wastage and yield loss [23]. Only appropriate planting density can effectively exert population advantages, optimize plant canopy structure, enhance leaf photosynthetic performance, increase the harvest index, and without increasing the risk of lodging [24,25]. Therefore, a rational planting density is a key agronomic practice for creating a reasonable population structure, enhancing the quality of crop stems, and ultimately increasing crop yield and quality.

In summary, both planting density and the stem physicochemical properties can influence lodging, but research that integrates these three aspects is limited. Therefore, in this study, we selected two *oat* varieties with different lodging resistance and combined them with different planting densities. We compared the differences in stem physicochemical properties between *oat* varieties with different lodging resistance. We also investigated the variations in physicochemical components of *oat* stems under different planting densities and explored the response mechanisms of lodging to density. Additionally, through a comprehensive analysis of the accumulation of stem physicochemical properties and the lodging resistance of plants at various planting densities, we identified the suitable planting density conducive to forage production. This study not only contributes to revealing the intrinsic connections between planting density and crop lodging, providing a theoretical basis for lodging control, but also offers practical strategies for improving forage quality.

## 2. Results

### 2.1. Differences Analysis between Two Varieties with Different Lodging Resistance

There were highly significant differences (*p* < 0.01) in the contents of calcium, potassium, silicon, crude fiber, lignin, starch, soluble sugar, and soluble protein between the lodging-resistant variety LENA and the lodging-sensitive variety QY2 (Figure 1). The contents of calcium, potassium, silicon, crude fiber, and lignin in LENA were significantly higher than those in QY2, while the contents of starch, soluble sugar, and soluble protein were significantly lower in LENA compared to QY2. There was no significant difference in magnesium content between the two varieties.

### 2.2. Changes of Oat Stem Physicochemical Properties and Lodging Coefficient under Different Planting Densities

In 2018 and 2019, as the planting density increased, the calcium, potassium, magnesium, crude fiber, and lignin contents in the second and third stem internodes of both varieties declined. The lodging coefficient exhibited an upward trend, but there was no significant difference in the lodging coefficient between varieties at planting densities of D2 and D3 (*p* < 0.05). The starch content of QY2 showed a trend of initially increasing and then decreasing during both 2018 and 2019, while LENA exhibited the same trend only in 2019. Meanwhile, the soluble sugar content increased initially and then decreased for both varieties, while the soluble protein content decreased initially, then increased, and finally decreased again, reaching its maximum at D3. The silicon content displayed no apparent change trend (Figure 2).

### 2.3. Effect of Planting Density and Variety on Stem Physicochemical Properties and Lodging Coefficient

Variance analysis revealed that the *oat* variety had an extremely significant influence on all characteristics except magnesium (*p* < 0.001), with the most pronounced impact on the soluble sugar content (F = 3826.877, *p* < 0.001, partial η^2^ = 0.992), followed by potassium content (F = 629.901, *p* < 0.001, partial η^2^ = 0.952). Planting density significantly influenced all characteristics except silicon (*p* < 0.001), with the highest impact on the soluble sugar content (F = 965.997, *p* < 0.001, partial η^2^ = 0.989). There was an interaction between planting density and variety concerning potassium, silicon, starch, soluble sugar, soluble protein, and lodging coefficient, with the most substantial effect observed on the lodging coefficient (F = 91.805, *p* < 0.001, partial η^2^ = 0.896) (Table 1). Therefore, a simple main effect analysis was conducted using the lodging coefficient as an example, revealing significant simple effects of variety at planting densities D3 and D4 (*p* < 0.05). Similarly, there were significant simple effects of density in the lodging-sensitive variety QY2 (*p* < 0.001) (Table 2).

### 2.4. Effect of Stem Physicochemical Properties on Lodging Coefficient

The random forest model was employed to identify the key influencing characteristics of the *oat* lodging coefficient (Figure 3). The results indicated that the contents of crude fiber, soluble sugar, magnesium, and starch in *oat* stems had a highly significant impact on the lodging coefficient (*p* < 0.01). Potassium and lignin significantly influenced the lodging coefficient (*p* < 0.05), while silicon, calcium, and soluble protein content did not demonstrate a significant impact. Correlation analysis revealed that crude fiber, magnesium, potassium, and lignin were highly significantly negatively correlated with the lodging coefficient (*p* < 0.01), whereas starch and soluble sugar contents exhibited an extremely significantly positive correlation with the lodging coefficient (*p* < 0.01). Additionally, there were interactions among crude fiber, soluble sugar, magnesium, starch, potassium, and lignin.

### 2.5. Relationships between Variety, Planting Density, Physicochemical Properties and Lodging Coefficient

Using a segmented structural equation model to analyze the relationships among variety, planting density, physicochemical properties, and lodging coefficient, the results indicated a good model fit (Fisher’s C = 2.614, *p* = 0.856) (Figure 4). Variety had a direct impact on the lodging coefficient. Additionally, variety indirectly influenced lodging coefficient by affecting crude fiber and soluble sugar contents, while planting density indirectly affected lodging coefficient by modulating crude fiber and starch contents. Potassium, crude fiber, starch, and soluble sugar all exhibited direct effects on lodging coefficient, with soluble sugar demonstrating the largest direct effect (path coefficient = 0.815). By summing the direct and indirect effects to calculate the total effect, it was found that soluble sugar had the largest total effect at 0.989. Therefore, planting density and variety primarily influence the lodging coefficient by affecting soluble sugar content.

### 2.6. Reasonable Planting Density for Oat Forage Production

The TOPSIS method was used to comprehensively evaluate *oat* forage quality and lodging under various planting densities (Figure 5). The highest composite score index of 0.55 was found at planting density D1. Planting densities D2 and D3 exhibited an identical composite score index of 0.51. There were no significant differences between the composite score indexes at planting densities of D1, D2, and D3, but all of them significantly differed from the composite score index of D4.

## 3. Discussion

### 3.1. Variety Differences and Density Gradient Variations in Oat Stem Traits

The stem traits of two *oat* varieties with distinct lodging resistance exhibited significant differences, suggesting that these traits are influenced not only by cultivation practices but also by inherent genetic characteristics [26]. Studies have shown that lodging-resistant varieties have higher levels of calcium, potassium, magnesium, silicon, lignin, and hemicellulose contents compared to lodging-sensitive varieties [27]. Our research had similar conclusions regarding the levels of calcium, potassium, silicon, crude fiber, and lignin. However, the magnesium content did not exhibit a significant difference in the two varieties studied here. Magnesium, as a component of chlorophyll, promotes photosynthesis and organic compound accumulation, thereby enhancing stem density and rigidity. From this perspective, lodging-resistant varieties would theoretically present higher magnesium content. Nevertheless, the transportation and transformation of substances within organisms are highly complex, and while individual elemental content can serve as an indicator of lodging resistance, it cannot be the sole criterion for evaluation.

The significantly higher contents of starch and soluble sugar in the stems of lodging-sensitive *oat* varieties, contrary to the findings of Deng et al. [28], can be attributed to the different varieties used in our study. Genetic differences among varieties can lead to varying research outcomes. A more reasonable explanation is that starch and soluble sugar, as crucial non-structural carbohydrates, primarily function in coordinating the source-sink relationship of crops under different growth stages and environmental conditions [29,30]. The accumulation of starch and soluble sugar in lodging-resistant varieties exceeds their developmental requirements, leading to their conversion into structural substances such as lignin and cellulose, thereby reducing their accumulation [31]. Furthermore, we observed that variety with high soluble protein content exhibited weaker lodging resistance, consistent with the findings of Croy and Hageman [32]. The soluble protein content can, to some extent, reflect the reduction amount of total nitrogen in plants and is a primary form of nitrogen existence in plant tissues. Higher nitrogen concentrations can inhibit lignin accumulation, resulting in weaker stems and reduced mechanical strength, consequently increasing the susceptibility to lodging [33].

Cultivation practices are crucial methods for regulating plant growth and development, with planting density receiving particular attention due to its simplicity and efficiency. Previous studies have shown that an increase in planting density is accompanied by a rise in the lodging rate [34]. This is the same as the result of our study. In addition, we found that stem lignin, crude fiber, potassium, calcium, and magnesium contents decreased with increasing planting density, which may be attributed to the decrease in effective radiation due to shading caused by high-density planting, thus affecting plant growth [35]. Starch and soluble sugar contents exhibited a trend of first increasing and then decreasing; a possible explanation is that optimal planting density fosters beneficial competition within the population, resulting in heightened starch and soluble sugar levels. Conversely, high planting density diminishes light transmittance within the canopy, leading to reduced net photosynthetic rates and accumulation of photosynthetic products [36]. It is noteworthy that soluble protein content showed a trend of initial decline followed by increase and then decrease with increasing planting density. As a vital component of life activities, soluble protein appears particularly sensitive to changes in the planting density.

### 3.2. Relationships among Variety, Planting Density, Stem Physicochemical Properties and Lodging Coefficient

Stem development is determined by a number of characteristics. This study found that both variety and planting density had a highly significant impact on the starch and soluble sugar contents in stems (*p* < 0.001, partial η^2^ > 0.90). This could be attributed to the fact that starch and soluble sugar, as major characteristics in maintaining plant development and responding to environmental regulation, are more sensitive to changes in the external environment. Moreover, starch and soluble sugar are key characteristics affecting the lodging coefficient. Prior research has indicated that increased deposition of starch and soluble sugar can enhance stem density and plumpness, consequently enhancing stem lodging resistance [37]. However, some studies have proposed that lodging is not directly associated with variations in carbohydrate content within stems, but instead, might be closely linked to their arrangement and interaction within the stem cell wall [38]. In our study, starch and soluble sugar showed a significant positive correlation with the lodging coefficient, indicating that elevated levels of starch and soluble sugar may increase the risk of lodging. This may be due to the fact that they have not been timely converted into structural substances and cannot effectively play a supporting role.

Crude fiber, comprising cellulose, hemicellulose, lignin, and cutin, is the primary component of plant cell walls. It stabilizes cell walls, enhances the mechanical strength and stiffness of stems, and plays a crucial role in crop lodging resistance [11]. Lignin, an essential constituent of crude fiber, provides rigidity to stems and is considered closely related to crop lodging resistance [39]. Planting density can regulate the synthesis of crude fiber and lignin, thereby influencing lodging. Previous studies have demonstrated that lower planting density can effectively enhance leaf photosynthetic activity, increase lignin synthesis-related enzyme activity, and accumulate carbohydrates in the stems, thereby enhancing the stem lodging resistance [40]. This study observed a significant influence of planting density on the content of crude fiber and lignin (*p* < 0.001). Further analysis revealed that both crude fiber and lignin were important influencing characteristics for the lodging coefficient. Therefore, we speculated that high planting density primarily increases the risk of lodging by inhibiting the accumulation of lignin and crude fiber. The structural equation model supported our speculation regarding crude fiber, while lignin is mainly determined by variety characteristics and indirectly influences the lodging coefficient.

Calcium, potassium, magnesium, and silicon are important mineral elements in crop stems, playing crucial roles in crop growth and stem rigidity maintenance. Our observations revealed that only silicon content did not respond significantly to density changes. Previous studies have indicated that calcium, potassium, magnesium, and silicon contents all exert negative effects on lodging index [41], suggesting that higher levels of these elements in stems can enhance crop lodging resistance. However, in our study, only potassium and magnesium contents significantly and negatively affected the lodging coefficient, especially potassium content, which had a direct and indirect impact on the lodging coefficient. The accumulation of silicon has been reported to increase stem cell wall thickness and vascular bundle size, thereby reducing the lodging index [42]. Calcium, as a crucial constituent of plant cell walls, contributes to their formation and strength enhancement [16]. Nonetheless, in our study, calcium and silicon contents did not significantly affect the lodging coefficient. These findings suggest that the relationship between mineral elements and crop lodging resistance may be more complex than previously perceived, warranting further research to comprehensively elucidate the underlying mechanisms.

### 3.3. Optimal Planting Density for Oat

Reasonable planting density can effectively regulate the forage quality and lodging of *oat*. Studies have indicated that optimal planting density can optimize plant growth, improve stem quality, and reduce lodging risk [43,44]. In our investigation, no significant differences were observed in the comprehensive score index of *oat* at planting densities D1, D2, and D3. However, lower planting density may lead to yield losses, as the increase in yield largely depends on the increase in planting density [45]. Therefore, higher planting density is preferable as long as it does not increase lodging or decrease forage quality. Hence, we have reason to believe that D3 is a reasonable planting density that can ensure both forage yield and quality.

In summary, the selection of planting density for *oat* production should strike a balance between yield, quality, and lodging resistance. While maximizing yield is important, it should not come at the cost of compromising forage quality or increasing the risk of lodging. Our result suggested that D3 represents a suitable compromise, offering a balance between these key characteristics. Future research could further explore the optimal planting density for specific *oat* varieties and growing conditions to achieve the best possible outcomes in terms of yield, quality, and lodging resistance.

## 4. Materials and Methods

### 4.1. Experimental Site

The experiment was conducted in Xining City, Qinghai Province, China (101°33′20″ E, 36°30′57″ N), with an average altitude of 2592 m above sea level and a plateau continental-type climate characterized by a cold and humid, but no absolute frost-free period. The average annual temperature was 5.1 °C, the average annual precipitation was 510 mm (mostly concentrated in July–September), and the average annual evaporation was 1830 mm. The air temperature and precipitation in 2018 and 2019 are shown in Figure 6 (http://tjj.qinghai.gov.cn/ (accessed on 20 September 2024)). 

### 4.2. Experimental Design and Field Management

The two *oat* varieties used in this study—lodging-resistant variety LENA and lodging-sensitive variety QY2—were obtained from the Qinghai Academy of Animal and Veterinary Science; the material characteristics are shown in Table 3. A two-year (2018 and 2019) field experiment was conducted using a split plot design, and the four planting densities used for this experiment were 2.25 × 10^6^ plants/ha (D1), 4.5 × 10^6^ plants/ha (D2), 6.75 × 10^6^ plants/ha (D3), and 9 × 10^6^ plants/ha (D4). The actual sowing quantity of each variety was calculated based on germination rate and 1000-grain weight. For LENA, the average 1000-grain weight was 27.77 g, the germination rate was 95%, and its sowing quantities were 65.7 kg/ha (D1), 131.6 kg/ha (D2), 197.3 kg/ha (D3), and 263.1 kg/ha (D4). In contrast, for QY2, the average 1000-grain weight was 32.20 g, the germination rate was 95%, and its sowing quantities were 76.2 kg/ha (D1), 152.6 kg/ha (D2), 228.8 kg/ha (D3), and 305.1 kg/ha (D4). Three plots were planted for each treatment; the area of each plot was 15 m^2^ (3 m × 5 m; *n* = 3), with row spacing of 20 cm and block spacing of 1 m. Furthermore, 150 kg/ha diammonium phosphate and 75 kg/ha urea were applied as base fertilizers before sowing, and plots were hand-weeded three times. The local production planting density is 6.75 × 10^6^ plants/ha.

### 4.3. Plant Sampling and Measurements

#### 4.3.1. Main Agronomic Traits and Lodging Coefficient

At the milk stage, 4 uniform plants were randomly selected from each plot (total 12 plants per planting density) to measure key agronomic traits, including plant height (PH: the distance from the plant base to the highest point at the top), root fresh weight (RFW: the fresh weight of the underground portion of the plant), and plant above-ground fresh weight (AFW: the fresh weight of the above-ground portion of the plant, consisting of stems, leaves, sheaths, and spikes).

Another set of 12 uniform plants were randomly selected from each planting-density plot. The YYD-1 strength tester (Zhejiang Top Technology Co., Ltd., Hangzhou, China) was used to measure the breaking strength (BS) of the second and third stem internodes. This was carried out as follows: The stalks (without leaf sheaths) were placed in the groove of the tester, with a distance of 2 cm between the two points. The bending probe was then vertically pressed downward into the middle of the internode at a constant speed. The maximum force that broke the stalk was recorded as the breaking strength. Then the PH, AFW, RFW, and BS were used to calculate the lodging coefficient (LC) as follows [46]:(1)$$LC=\frac{PH\times AFW}{RFW\times BS}$$
where BS is the mean value of breaking strength between the second and third stem internodes.

#### 4.3.2. Physicochemical Indicators 

*Oat* plants at the milk stage were selected, and the second and third stem internodes above the ground were first oven-dried at 105 °C for 30 min and then at 65 °C to constant weight. Thereafter, they were ground and passed through a 60-mesh sieve for the determination of physicochemical properties contents.

##### Determination of Mineral Element Content

The mineral element content was determined using the method described as follows [47,48]. Approximately 0.2 g of dried *oat* stem powder was weighed into a digestion tube, and 5 mL of HNO_3_ and 4 mL of H_2_O_2_ solution were added. Then, the mixture was digested in a Mars6-Xpress microwave digestion system (CEM Technology Co., Ltd., Matthews, NC, USA) for 30 min. After complete digestion, the digestion tube was transferred to an XMTG-7000 temperature controller (Gongsheng Instrument Co., Ltd., Yuyao, China), and the acid was evaporated at 120 °C until no yellow smoke emerged. Subsequently, the solution in the digestion tube was transferred to a 50 mL volumetric flask and diluted to 50 mL with ultrapure water. Calcium (Ca), potassium (K), magnesium (Mg), and silicon (Si) contents were determined using an ICPE-9000 inductively coupled plasma emission spectrometer (Shimadzu Corporation Co., Ltd., Kyoto, Japan); the operating parameters are shown in Table 4.

##### Determination of Crude Fiber and Lignin Contents

Crude fiber (CF) content (%) was determined using the method described by Zakirullah et al. [49]. We took approximately 1.0 g of the sample (W1) in a 250 mL beaker and added 1.25% H_2_SO_4_ to make the volume up to 200 mL. The mixture was digested by micro-boiling for 30 min, and then filtered and washed. Subsequently, we added 1.25% NaOH and made up the volume up to 200 mL. Then, we heated the mixture for 30 min and filtered and washed the residue. This residue was placed in a pre-weighed crucible and then in an oven at 105 °C for 24 h for drying. After recording the dry weight (W2), the sample was placed in a muffle furnace at 600 °C for 4 h and weighed after cooling (W3). Finally, the following formula was used to calculate the crude fiber content: Crude fiber (%) = (W2 − W3)/W1 × 100(2)

Lignin (LI) content (%) was determined using the method described by Brinkmann et al. [50]. We used approximately 0.5 g of the sample (W1) in a 250 mL beaker, added 100 mL of 0.5 M H_2_SO_4_ (containing 1 g of cetyltrimethylammonium bromide), and boiled the mixture for 1 h under continuous stirring. A drop of octan-2-ol was added as an antifoam agent. We filtered and washed the mixture 3–5 times with distilled water and then washed it with acetone until further decoloration was not observed. The residue was dried at 105 °C for 2 h, followed by the addition and mixing of 10 mL of 72% H_2_SO_4_ and then another 10 mL of 72% H_2_SO_4_ after 1 h for continued hydrolysis for 3 h. The residue was then washed with distilled water until it was acid free, dried at 105 °C for 2 h, cooled, and weighed (W2). The residue was placed in a muffle furnace at 500 °C for 3 h, cooled, and weighed again to determine ash content (W3). Lignin content was then calculated as follows:Lignin (%) = (W2 − W3)/W1 × 100(3)

##### Determination of Starch and Soluble Sugar Contents

The starch (ST) and soluble sugar (SS) were determined following the procedure outlined by Zhang et al. [51]. Specifically, 0.1 g of dried *oat* stem powder was weighed and 2 mL of 80% ethanol was added. The mixture was incubated in an 80 °C water bath for 30 min. After cooling, it was centrifuged at 4000× *g* for 10 min, and the supernatant was collected. This extraction process was repeated 3 times, and all the supernatants were combined as the soluble sugar extract. A 100 μL aliquot of the extract was mixed with 50 μL of anthrone-ethyl acetate solution and 500 μL of H_2_SO_4_. Following shaking, the mixture was boiled for 1 min, cooled, and the absorbance was measured at 630 nm.

For the residue after removing soluble sugar, 2 mL of distilled water was added, mixed, and then gelatinized in a boiling water bath for 15 min. Afterward, 2 mL of 9.2 mol/L perchloric acid was added, shaken for 15 min, and 4 mL of distilled water was added. The mixture was centrifuged at 4000× *g* for 10 min, and the supernatant was collected. To the precipitate, another 2 mL of 4.6 mol/L perchloric acid was added, and the extraction process was repeated. The supernatants were combined to determine the starch content. A 50 μL aliquot of the extract was mixed with 150 μL of distilled water and 1 mL of sulfuric acid–anthrone solution, boiled for 2 min, cooled, and the absorbance was measured at 620 nm.

##### Determination of Soluble Protein Content

The soluble protein (SP) was determined as follows [52,53]. One hundred and fifty milligrams of *oat* stem powder was accurately weighed, and 10 mL of extraction buffer (0.7 M Sucrose, 50 mM EDTA, 0.1 M KCl, 10 mM Thiourea, 0.5 mM Tris, 2 mM PMSF, 50 mM DTT) was added. The mixture was then incubated on ice for 20 min at 4 °C and centrifuged at 14,000× *g* for 15 min. An appropriate volume of supernatant was transferred to a test tube, mixed with 5 mL of Coomassie Brilliant Blue G-250 staining solution, incubated for 5 min, and the absorbance was measured at 595 nm.

### 4.4. Data Analyses

The data were analyzed using Microsoft Excel 2010 (Microsoft, Redmond, WA, USA) and SPSS Statistics 22.0 (IBM, Armonk, NY, USA). Duncan’s test (*p* < 0.05) was applied to compare the significance of characteristic means; an independent samples *t*-test was used to compare differences between two varieties (*p* < 0.05); and analysis of variance (ANOVA) using a general linear model was performed. Additionally, a random forest model was constructed using the “Random Forest” package, the segmented structural equation model was constructed using the “Piecewise SEM” package, and TOPSIS analysis was performed using the “TOPSIS” package in R 4.3.2 (R Development Core Team, Vienna, Austria, 2023). OriginPro 2021 (Origin Lab, Northampton, MA, USA) was used for generating graphs.

## 5. Conclusions

Higher levels of calcium, potassium, silicon, crude fiber, and lignin help to improve *oat* lodging resistance, along with lower levels of soluble sugar, starch, and soluble protein. Crude fiber, soluble sugar, magnesium, starch, potassium, and lignin are the key characteristics affecting the lodging coefficient, and planting density also has a significant effect on these characteristics (*p* < 0.01). Variety and planting density affect the lodging coefficient mainly by regulating the synthesis of starch, soluble sugar, and crude fiber. Among the investigated planting densities, D3 emerges as the most conducive to nutrient accumulation, with lodging risk remaining within an acceptable range, thereby rendering it the optimal planting density for *oat* forage production.

## Figures and Tables

**Figure 1 plants-13-02739-f001:**
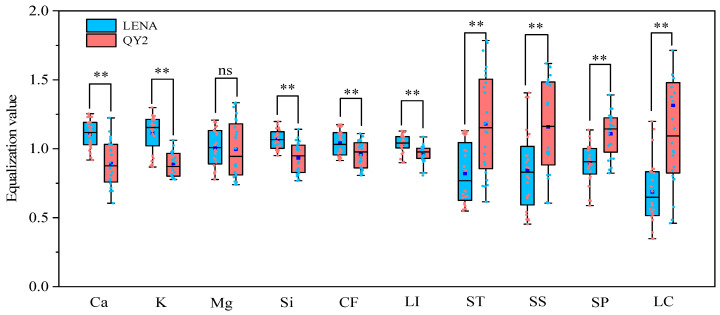
Differences in physicochemical properties and lodging coefficient between two varieties. Note: Ca, calcium; K, potassium; Mg, magnesium; Si, silicon; CF, crude fiber; LI, lignin; ST, starch; SS, soluble sugar; SP, soluble protein; LC, lodging coefficient. ** denotes significant differences at *p* < 0.01; ns represents not significant (comparing using independent sample *t*-test).

**Figure 2 plants-13-02739-f002:**
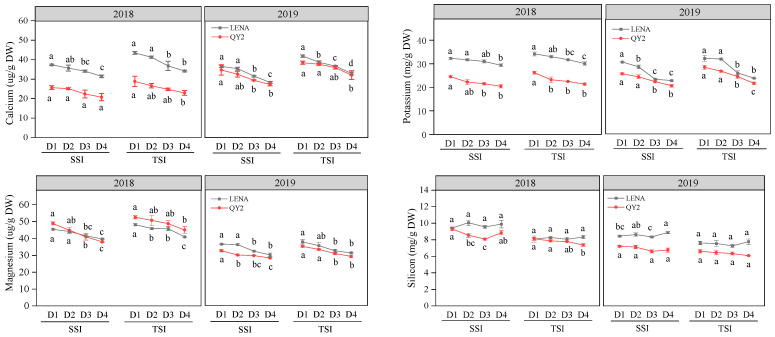
Changes of *oat* stem physicochemical properties and lodging coefficient under different planting densities. Note: D1, D2, D3, and D4 represent different planting densities. SSI, second stem internode; TSI, third stem internode. The results are expressed as mean value ± SEM. Different letters represent significant differences at *p* < 0.05 (comparing using Duncan’s test).

**Figure 3 plants-13-02739-f003:**
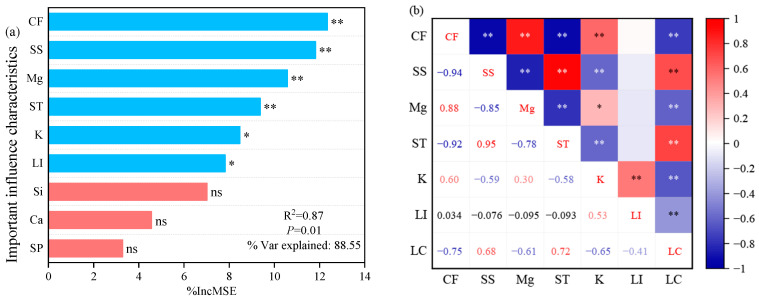
Key characteristics affecting the lodging coefficient and their relationships. Note: CF, crude fiber; SS, soluble sugar; Mg, magnesium; ST, starch; K, potassium; LI, lignin; Si, silicon; Ca, calcium; SP, soluble protein; LC, lodging coefficient. * and ** denote significant differences at *p* < 0.05 and *p* < 0.01, respectively; ns represents not significant. The blue color represents a significant impact and the red color represents a non-significant impact in Figure (**a**). The numbers represent the correlation coefficients in Figure (**b**).

**Figure 4 plants-13-02739-f004:**
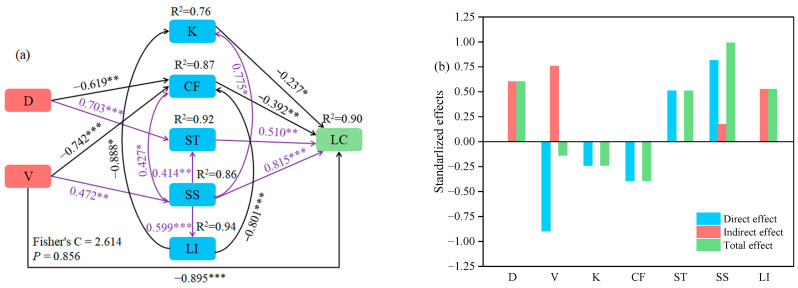
Paths of different varieties and planting densities on the lodging coefficient of *oat* (**a**) and standardized effect values for each variable (**b**). Note: D, planting density; V, variety; K, potassium; CF, crude fiber; ST, starch; SS, soluble sugar; LI, lignin; LC, lodging coefficient. In figure (**a**), the black and purple arrows indicate negative and positive path relationships, respectively, and the numbers indicate normalized path coefficients. *, ** and *** denote significant differences at *p* < 0.05, *p* < 0.01 and *p* < 0.001, respectively.

**Figure 5 plants-13-02739-f005:**
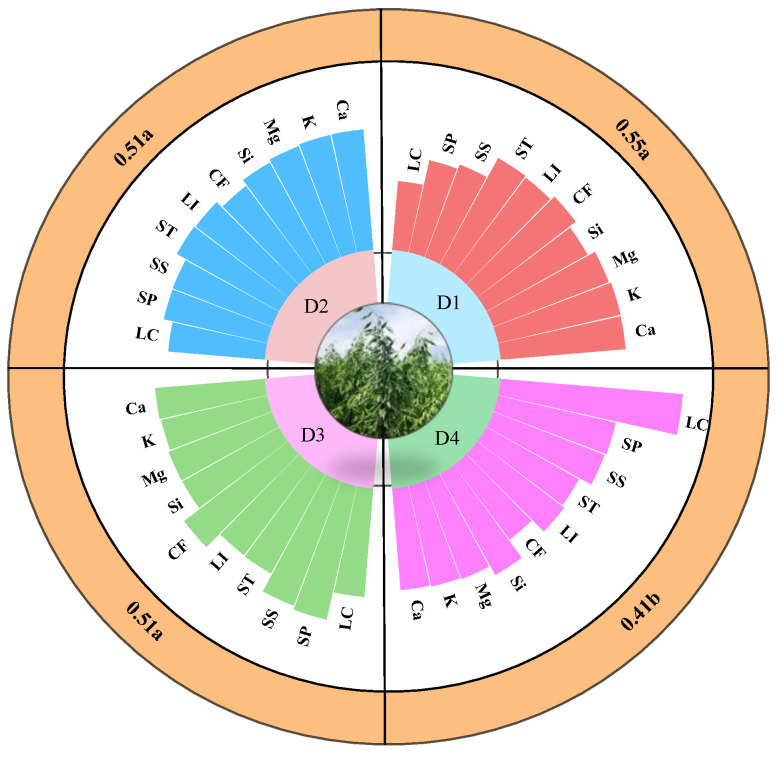
A comprehensive analysis of planting densities that favor *oat* forage production. Note: Ca, calcium; K, potassium; Mg, magnesium; Si, silicon; CF, crude fiber; LI, lignin; ST, starch; SS, soluble sugar; SP, soluble protein; LC, lodging coefficient. The outer ring of the figures represents the comprehensive score indexes under four planting densities, and the letters indicate significant differences (*p* < 0.05) between the comprehensive score indexes (comparing using Duncan’s test). Different colors represent different planting densities.

**Figure 6 plants-13-02739-f006:**
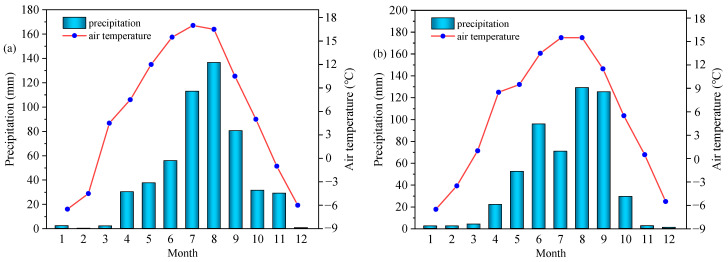
Air temperature and precipitation for 2018 (**a**) and 2019 (**b**).

**Table 1 plants-13-02739-t001:** Effect of planting density and variety on stem physicochemical properties and lodging coefficient.

Characteristics	Variety Main Effect			Density Main Effect			Variety * Density		
	F	*p*	Partial η^2^	F	*p*	Partial η^2^	F	*p*	Partial η^2^
Ca	277.674	<0.001	0.897	55.920	<0.001	0.840	0.981	0.414	0.084
K	629.901	<0.001	0.952	103.738	<0.001	0.907	3.005	0.045	0.220
Mg	0.589	0.449	0.018	45.487	<0.001	0.810	0.266	0.849	0.024
Si	176.040	<0.001	0.846	3.720	0.021	0.259	5.151	0.005	0.326
CF	347.593	<0.001	0.916	75.573	<0.001	0.876	0.978	0.415	0.084
LI	132.930	<0.001	0.806	75.938	<0.001	0.877	0.288	0.834	0.026
ST	1067.410	<0.001	0.971	137.716	<0.001	0.928	37.316	<0.001	0.778
SS	3826.877	<0.001	0.992	965.997	<0.001	0.989	67.080	<0.001	0.863
SP	436.853	<0.001	0.932	206.239	<0.001	0.951	6.483	0.001	0.378
LC	571.908	<0.001	0.947	267.687	<0.001	0.962	91.805	<0.001	0.896

Note: Ca, calcium; K, potassium; Mg, magnesium; Si, silicon; CF, crude fiber; LI, lignin; ST, starch; SS, soluble sugar; SP, soluble protein; LC, lodging coefficient. * represents the interaction between variety and density.

**Table 2 plants-13-02739-t002:** Simple main effects analysis of the lodging coefficient.

D	Simple Effects of Variety			V	Simple Effects of Density		
	F	*p*	Partial η^2^		F	*p*	Partial η^2^
D1	1.269	0.267	0.031	LENA	1.382	0.262	0.094
D2	2.874	0.098	0.067	QY2	18.747	<0.001	0.584
D3	5.519	0.024	0.121				
D4	37.784	<0.001	0.486				

Note: D represents the planting density, V represents variety.

**Table 3 plants-13-02739-t003:** Characteristics of tested materials.

Variety	Characteristics
LENA	Plant height 100–115 cm, medium-late maturity, panicle type is scattered, 2–5 tillers
QY2	Plant height 120–140 cm, early maturity, panicle type is scattered, 2–3 tillers

**Table 4 plants-13-02739-t004:** I CPE-9000 operating parameters.

Parameter	Condition
Forward power	1.2 kW
Plasma gas flow rate	10 L/min
Auxiliary gas flow rate	0.6 L/min
Carrier gas flow rate	0.7 L/min
Instrument stabilization delay	15 s

## Data Availability

The original contributions presented in the study are included in the article, further inquiries can be directed to the corresponding author.

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
