# Peer review of "Differences in Physicochemical Properties of Stems in Oat (*Avena sativa* L.) Varieties with Distinct Lodging Resistance and Their Regulation of Lodging at Different Planting Densities"

_plants, 2024, doi:10.3390/plants13192739_

Round 1

Reviewer 1 Report

Comments and Suggestions for Authors

Specific comments:

1) The iThenticate report of 38% need to further reduce by the authors to ≤ 10%.

2) Authors should follow the journal guidelines by revising the abstract to maximum of 200 words.

3) The justification of the research should be clearly stated in the Introduction section.

4) Figures 1, 2, and 4b only the vertical and horizontal lines should show.

5) The discussion can be improved with reference to the recent citations relating to the study.

Minor comments.

L2-3: Italicize the Avena sativa

L404: Check appropriate style of citation, et al. Check throughout the MS.

Author Response

Comments 1: The iThenticate report of 38% need to further reduce by the authors to ≤ 10%.

Response 1: Thanks to the reviewers' comment. We have checked the full article and can ensure that there is no plagiarism. Also, most of the methods used in the article are quoted from others, which could be the main reason for the high repetition rate, but this really cannot be avoided.

Comments 2: Authors should follow the journal guidelines by revising the abstract to maximum of 200 words.

Response 2: Thank you for pointing this out. We agree with this comment. Therefore, we have shortened the abstract section, the major modifications are marked in green.

Comments 3: The justification of the research should be clearly stated in the Introduction section.

Response 3: We thank the reviewers' valuable comment. In the Introduction, the first four paragraphs are devoted to the justification of the research. For example, the first paragraph writes about the importance of oat, the second paragraph writes about the problems of oat in production (lodging), the third and fourth paragraphs write about the ways through which the problem of lodging can be solved, and the last paragraph summarizes the objectives and significance of the research to be conducted by combining the factors mentioned above.

Comments 4: Figures 1, 2, and 4b only the vertical and horizontal lines should show.

Response 4: Thanks to the reviewers' comments. We checked all the figures in the article and compared them with those of published articles in plants journals, and concluded that there were no problems with our figures.

Comments 5: The discussion can be improved with reference to the recent citations relating to the study.

Response 5: Thanks to the reviewers' valuable comment. We have tried our best to cite the most recent literature in the discussion section, but we can't guarantee that all the literature is from the last few years because we need to consider the applicability of the references when citing.

Comments 6: L2-3: Italicize the Avena sativa

Response 6: Thank you for pointing this out. We have modified this error in the header and highlighted it in green.

Comments 7: L404: Check appropriate style of citation, et al. Check throughout the MS.

Response 7: We sincerely appreciate the valuable feedback from the reviewer. We checked the citation style in the article and found the same citation style in plants already published articles [1]. Therefore, we believe that this citation style is reasonable.

[1] Li, Q.; Zhao, X.; Wu, J.; Shou, H.; Wang, W. The F-Box Protein TaFBA1 Positively Regulates Drought Resistance and Yield Traits in Wheat. Plants 2024, 13, 2588. https://doi.org/10.3390/plants13182588

Reviewer 2 Report

Comments and Suggestions for Authors

All considerations (corrections and suggestions) are noted in the attached copy of this manuscript. The work may be published without problems, taking into account the questions raised.

Author Response

Comments 1: To inform in this abstract where and when this research was carried out.

Response 1: Thank you for pointing this out. We agree with this comment. We have added the duration of the study in the abstract section, but due to word limitations, the trial site will not be reflected in the abstract section, but reflected in 4.1 (Experimental site). Modifications are highlighted in red.

Comments 2: Instead of "we utilized" replace with it was used. in scientific language always put in the impersonal.

Response 2: Agree. We have made changes to the sentences in conjunction with comment 1, which are highlighted in red in the abstract section.

Comments 3: Instead "our" replace in this study...

Response 3: Agree, and we have deleted the entire sentence in conjunction with other reviewers' comments.

Comments 4: Do not repeat the same words that exist in the manuscript title in keywords. include other terms.

Response 4: Agree, and we deleted words from the keywords that appear in the title.

Comments 5: Does this last sentence of the introduction correspond to the general objective of the work? note that the conclusions of this research will be the answers to these objectives. it is important to make this consistency clear.

Response 5: We have carefully checked the article and believe that there is no problem with this sentence, which states the significance of this study, and that the objectives of this study are reflected earlier in this sentence.

Comments 6: Teste envolvi do "F ou t tes" da ANOVA ?

Response 6: We used independent sample t-test. We have added notes and highlighted them in red in the Notes section of Figure 1.

Comments 7: Which mean comparison test was performed (Duncan's test) P=0.05?

The comparisons of envalidated means admit that all variables showed significant interaction of factors (Varieties x Densities), which is not true.

Response 7: Yes, we used Duncan's test and added the relevant content in the notes section of Figure 2. Figure 2 demonstrates the trend of various physicochemical properties at different planting densities, and the lower case letters in the figure indicate the differences of various physicochemical properties at different planting densities, and do not reflect the interactions between varieties and densities. Of course, we show this part in Table 1, not all factors show significant interaction relationship, as can be seen from Table 1, the effect of variety and planting density interaction on the content of calcium, magnesium, crude fiber and lignin is not significant.

Comments 8: Replace "factors" with Variables or characteristics.

Response 8: Thanks to the reviewers' valuable comment. We agree with this comment. We have checked and changed the article by replacing “factors” with “characteristics”, and the changes highlighted in red.

Comments 9: What mean comparison test was performed (Duncan's test) P=0.05?

Response 9: Yes, we used Duncan's test and added to it in the notes section of Figure 5.

Comments 10: The number of blocks involved in the experiment (4 or 5 blocks?) in subdivided plots was not defined.

Response 10: Agree, We have made additions in section 4.2 (Experimental design and field management), and highlighted it in red.

Comments 11: Conclusions should be written directly, in short sentences without explanation. conclusions explain why they are transformed into results. it is important to rearrange the conclusions and use the verb in the present tense. see for example this first conclusion below: Higher levels of calcium, silicon, crude fiber and lignin interfere with lodging resistance, along with lower levels of soluble sugar, starch and soluble protein.

Response 11: Thanks to the reviewers' valuable comment. Based on reviewers' comment, we have made changes to the conclusions section and highlighted them in red.

Reviewer 3 Report

Comments and Suggestions for Authors

The manuscript is relevant and generally well written. It deals with a study combining varieties and oat planting density, seeking to better understand and avoid lodging. The methodology was based on physical-chemical and lodging analyses. The results provide information that explains the combination of factors involved and can be applied in practice immediately.

In the discussion regarding soluble protein content, there was an initial downward trend followed by an increase and decrease with increasing planting density. The authors state that as a vital component of vital activities, soluble protein appears to be particularly sensitive to changes in the external environment. In this case, to make this statement clearer, the authors should mention the environmental conditions (climate factors) during the experiment and explain how these factors led to the result found.

In the study, only potassium and magnesium levels significantly and negatively affected the lodging coefficient, especially potassium content, which had a direct and indirect impact on the lodging coefficient, while Ca and Si did not, contrary to other studies. The authors suggest that the relationship between mineral elements and resistance to crop lodging may be more complex than previously thought, which justifies additional research to comprehensively elucidate the underlying mechanisms. However, they do not mention in the slightest what research would be necessary for this elucidation.

The authors suggest that planting density D3 is the most appropriate, however, here is a better discussion of the reasons for this statement. Is this density close to what is done in practice?

Figure 2 has several parts that are small, making it difficult to understand. Perhaps increasing each variable for the years 2018 and 2019 will resolve this issue.

The same occurs for Figure 4.

Author Response

Comments 1: The manuscript is relevant and generally well written. It deals with a study combining varieties and oat planting density, seeking to better understand and avoid lodging. The methodology was based on physical-chemical and lodging analyses. The results provide information that explains the combination of factors involved and can be applied in practice immediately.

Response 1: We sincerely appreciate the recognition and efforts of the reviewer in evaluating our work.

Comments 2: In the discussion regarding soluble protein content, there was an initial downward trend followed by an increase and decrease with increasing planting density. The authors state that as a vital component of vital activities, soluble protein appears to be particularly sensitive to changes in the external environment. In this case, to make this statement clearer, the authors should mention the environmental conditions (climate factors) during the experiment and explain how these factors led to the result found.

Response 2: Thanks to the reviewers' valuable comment. For all treatments, the climatic conditions were the same, so the “change in external environment” here refers to the “change in planting density”. Compared with other physicochemical components, the response of soluble protein content to planting density is more complex, so it is presumed that proteins are more sensitive to changes in the external environment. To make the expression clearer, we have modified the last paragraph of 3.1 (Variety differences and density gradient variations in oat stem traits) and marked it with purple color. We have not yet found a reasonable explanation for the reason for this phenomenon, and we will continue to explore it in subsequent studies.

Comments 3: In the study, only potassium and magnesium levels significantly and negatively affected the lodging coefficient, especially potassium content, which had a direct and indirect impact on the lodging coefficient, while Ca and Si did not, contrary to other studies. The authors suggest that the relationship between mineral elements and resistance to crop lodging may be more complex than previously thought, which justifies additional research to comprehensively elucidate the underlying mechanisms. However, they do not mention in the slightest what research would be necessary for this elucidation.

Response 3: Thank you for the valuable feedback from the reviewer. Our research has shown a different phenomenon from previous studies, which may be due to various reasons. We believe that more research should be conducted to clarify its mechanismIt also provides a direction for our future research, but the specific research to be conducted is not within the scope of this article.

Comments 4: The authors suggest that planting density D3 is the most appropriate, however, here is a better discussion of the reasons for this statement. Is this density close to what is done in practice?

Response 4: Yes, it is close to the actual production situation. We supplemented 4.2 (Experimental design and field management) with actual planting densities for local production, highlighted in purple.

Comments 5: Figure 2 has several parts that are small, making it difficult to understand. Perhaps increasing each variable for the years 2018 and 2019 will resolve this issue.

The same occurs for Figure 4.

Response 5: Thanks to the reviewers' comment, but we think that the figure is not a problem, Figure 2 shows the changes in the physicochemical properties of the stems of the two varieties at different planting densities in 2018 and 2019, for the meanings indicated by the letters in the figure, we have also explained them in the notes section under the figure, and we think that the figure in the article is able to clearly respond to the objectives of the study.

Figure 4 shows the analysis done by combining two years of data together, which we think is more convincing for the problem we are trying to solve. In addition, to be able to make the expression clearer, we added an explanation of the figures in the notes section and highlighted them in purple.

Reviewer 4 Report

Comments and Suggestions for Authors

The subject of the article presented for review is very interesting. The study used new analyses. The problem of oat lodging is very big, sowing density and correct variety selection have a big impact on lodging. Unfortunately, in addition to the above-presented facts, thermal and humidity conditions have a big impact on oat lodging. The article lacks data on precipitation and temperature. In addition, the authors have results from only one year of research. Variable conditions in at least two years will allow for drawing correct conclusions. Unfortunately, in its current form, the article is not suitable for publication in my opinion.

My decision is to reject

Author Response

Comments 1: The subject of the article presented for review is very interesting. The study used new analyses. The problem of oat lodging is very big, sowing density and correct variety selection have a big impact on lodging. Unfortunately, in addition to the above-presented facts, thermal and humidity conditions have a big impact on oat lodging. The article lacks data on precipitation and temperature. In addition, the authors have results from only one year of research. Variable conditions in at least two years will allow for drawing correct conclusions. Unfortunately, in its current form, the article is not suitable for publication in my opinion.

Response 1: Many thanks to the reviewers' valuable comment. We strongly agree that thermal and humidity have a great influence on lodging, so according to the reviewer's request, we have added information about precipitation and temperature in 2018 and 2019 in the 4.1 (Experimental site). In addition, there are many other factors (wind, soil type, fertilizer application, etc.) that affect lodging, but it is impossible to study all the factors affecting lodging in one article, so in this study, we chose the most common varieties (genetic factors) and planting density (cultivation measures) for our research, and we will continue to explore the other influencing factors in the subsequent trials.

Furthermore, our trials were conducted over two years in 2018 and 2019. This is reflected in the abstract, Figure 2, and in section 4.2 (Experimental design and field management).

Reviewer 5 Report

Comments and Suggestions for Authors

Lodging resistance is a key parameter in oat production because early lodging can cause serious yield losses and poor quality and lodging before the harvest makes this process more time-consuming and generates losses. There are lodging-resistant varieties, therefore, the appropriate variety selection would be the base of an efficient production technology. Besides the properties of genotypes and the plant density, the cultivation technological parameters such as fertilization and water supply can impact lodging significantly.

Specific comments:

- Please add some information regarding the habits of the tested genotypes (plant height, tillering ability, etc.).

- Please indicate in Figure 3 which correlation coefficients are statistically significant.

- Please add references to confirm the reality of the climatological description of the study area.

- It must be taken into account that specification of the growing season's weather conditions would be more informative than a general climatological specification.

- The authors could increase the practical aspects of the manuscript if the effects of the treatment on the forage yield were examined in a more detailed way.

Author Response

Comments 1: Lodging resistance is a key parameter in oat production because early lodging can cause serious yield losses and poor quality and lodging before the harvest makes this process more time-consuming and generates losses. There are lodging-resistant varieties, therefore, the appropriate variety selection would be the base of an efficient production technology. Besides the properties of genotypes and the plant density, the cultivation technological parameters such as fertilization and water supply can impact lodging significantly.

Response 1: We thank the reviewers' valuable comment. Lodging is indeed influenced by a number of factors, and in this study we investigated the effects of variety and planting density, and we will continue to investigate in depth the effects of other factors (fertilization, water, etc.) on lodging in the next studies.

Comments 2: Please add some information regarding the habits of the tested genotypes (plant height, tillering ability, etc.).

Response 2: Thank you for pointing this out. We agree with this comment. We have added a description of the tested material in section 4.2 (Experimental design and field management) and highlighted it in yellow (table 3).

Comments 3: Please indicate in Figure 3 which correlation coefficients are statistically significant.

Response 3: In the correlation heatmap in Figure 3, the lower left part is the correlation coefficient, and the “*” in the upper right part is its corresponding significance level, and those marked with “*”are statistically significant.

Comments 4: Please add references to confirm the reality of the climatological description of the study area.

Response 4: Agree. We have added the data access URL in section 4.1 (Experimental site) and highlighted it in yellow.

Comments 5: It must be taken into account that specification of the growing season's weather conditions would be more informative than a general climatological specification.

Response 5: Agree. We have added climate data for 2018 and 2019 in Section 4.1(Experimental site), which provides a clear response to temperature and precipitation during the growing season.

Comments 6: The authors could increase the practical aspects of the manuscript if the effects of the treatment on the forage yield were examined in a more detailed way.

Response 6: Thanks to the reviewers' valuable comment. It is indeed very important and practical to clarify the effect of treatments on yield, but this is not the focus of this study, and adding this section to this paper is a bit off-topic. Therefore, we have decided not to add anything about yield in this paper, but will specifically explore the effect of treatments on yield in other articles.

Round 2

Reviewer 4 Report

Comments and Suggestions for Authors

After adding data about years of research, temperature, etc. the article looks much better.

I propose accepting the article for publication